# Safe Robot Learning in Assistive Devices through Neural Network Repair

**Keyvan Majd**[1†], **Geoffrey Clark**[1], **Tanmay Khandait**[1], **Siyu Zhou**[1],
**Sriram Sankaranarayanan**[2], **Georgios Fainekos**[3], **Heni Ben Amor**[1]

[1]Arizona State University, [2]University of Colorado Boulder, [3] Toyota NA-R&D

[†]majd@asu.edu

**Abstract:** Assistive robotic devices are a particularly promising field of application for neural networks (NN) due to the need for personalization and hard-to-model human-machine interaction dynamics. However, NN based estimators and controllers may produce potentially unsafe outputs over previously unseen data points. In this paper, we introduce an algorithm for updating NN control policies to satisfy a given set of formal safety constraints, while also optimizing the original loss function. Given a set of mixed-integer linear constraints, we define the NN repair problem as a Mixed Integer Quadratic Program (MIQP). In extensive experiments, we demonstrate the efficacy of our repair method in generating safe policies for a lower-leg prosthesis.

**Keywords:** Imitation Learning, Assistive Robotics, Safety, Prosthesis

## 1 Introduction

Robot learning has the potential to revolutionize the field of wearable robotic devices, such as prosthetics, orthoses and exoskeletons [1]. Often, such devices are designed in a "one size fits all" manner using models based on average population statistics. However, machine learning techniques can help adapt control parameters automatically to the wearer's individual characteristics, motion patterns or biomechanics. The result is a substantial improvement in ergonomic comfort and quality-of-life. Despite these benefits, the adoption of machine learning and, in particular, deep learning [2] in this field is still limited, largely due to safety concerns. In the case of a prosthesis, for example, it may be important to guarantee that the generated control values do not exceed a maximum threshold under a set of testing conditions. Other safety constraints include limits on velocities, joint angles and bounds on the change in control inputs.

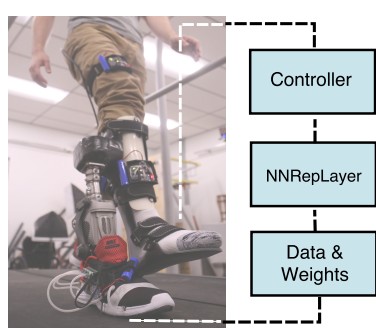

Figure 1: A lower-leg prosthesis running a neural network for control. A neural network repair process ensures that safety constraints are satisfied.

In this paper, we derive an algorithm for training neural networks controllers to satisfy given safety specifications, in addition to fitting the given training/test data. In particular, we use our approach to derive controllers for a robotic lower-leg prosthesis that satisfy basic safety conditions, (see Fig. 1). Our proposed approach inputs a trained network (e.g., a network obtained using backpropagation on the training data) along with a specification that places restrictions on the possible outputs for a given set of inputs. It then generates a modified set of weights that obeys the desired safety constraints on the output using deterministic global optimization. We provide theoretical guarantees of optimality for this technique (assuming no numerical errors). Furthermore, in real-world robot experiments we show that the introduced methodology produces safe neural policies for a lower-leg prosthesis satisfying a variety of constraints.

6th Conference on Robot Learning (CoRL 2022), Auckland, New Zealand.

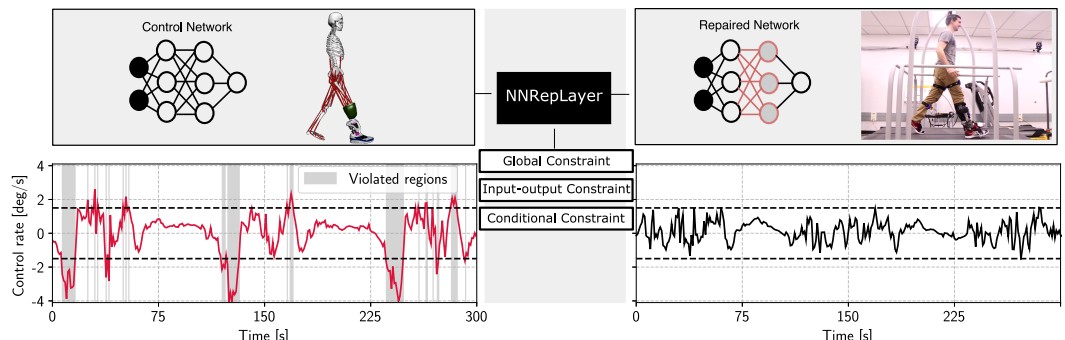

Figure 2: Overview of our approach. Left: a given (unsafe) neural network is trained to control a prosthesis. However, its outputs violate the formal safety constraints. Right: Using our NN Repair strategy, we identify an adjusted set of neural network weights that removes all violations while still maintaining the underlying behavior of the controller.

**Contributions.** The proposed method can repair any layer in a network increasing the size of the solution space and the likelihood of a feasible successful repair. More importantly, it comes with theoretical guarantees on successfully repairing all discovered unsafe samples. When compared to retraining or fine-tuning methods, it also has two distinct benefits: (1) it does not require modification of the training data so as to satisfy the constraints, and (2) it does not utilize gradient optimization methods which do not guarantee constraint satisfaction even for the discovered unsafe data points. Finally, when compared to other property driven repair works, i.e., [3, 4, 5, 6, 7], it is applied for the first time to a real physical system, i.e., a powered prosthetic device, as opposed to a model.

**Related Works.** Learning based control for prosthetics is motivated by the challenges in modeling human-prosthesis dynamics, which exhibits time varying behavior. Several different approaches that utilize learning based methods for controlling prosthetic devices have been proposed in the literature [8, 9]. Nevertheless, this short review focuses only on neural network (NN) based methods. Gao et al. [10] provide a prosthetic controller based on recurrent neural networks, and show that this type of controller effectively minimizes the difference between desired and actual trajectories on a powered prosthetic device. In another direction, Keleş and Yucesoy [11] and Vonsevych et al. [12] focus on utilizing electromyography (EMG) signals to predict control parameters for prosthetic ankles and hands. Our work follows a similar approach in predicting ankle control parameters with a neural network, which then drives the prosthesis through a PD controller. The verification of neural networks has been widely studied in order to check specifications for a given NN as a standalone component [13, 14, 15, 16] (for a survey see [17]). or as part of a closed loop system [18, 19, 20]. Testing techniques can produce counterexamples (adversarial samples) for a variety of NN-based applications, e.g., [21, 14, 22, 23, 24, 25]. One approach to ensuring that a NN satisfies given set of safety properties is to use retraining and fine-tuning based on counter-examples [26, 27, 7]. However, this approach has a number of pitfalls. First, the labels for the counterexamples need to be available. This may involve further data-collection, which is often cumbersome. Also, gradient descent optimization approaches cannot provide any guarantees that the result satisfies the provided constraints. Our approach, in contrast, avoids generating adversarial examples or performing gradient descent. The problem of "repairing" a network, which involves modifying weights of the network in a minimal manner to satisfy some safety properties, is thus more involved than simply retraining a network on some additional data involving counterexamples. Some methods in the literature [3, 4, 5, 6] make progress toward the goal of repairing NNs. Goldberger et al. [3] can only repair the output layer, which drastically reduces the space of possible successful repairs (and in many cases a repair is not even possible). The method of Fu and Li [4] produces patches for certain partitions of the input space wherein the corresponding outputs are modified. The direction of adding patches on the NN output is very promising; however, it is unclear how the approach will scale for high dimensional inputs, since it requires partitioning the input space into affine subregions. Authors in [6] repair the linear regions related to each faulty sample in the faulty network's weight space using a decoupled DNN architecture. However, this method causes the repaired network to be discontinuous. Therefore, it cannot be employed in robot learning and control applications which is the target application of this paper. This method is not also applicable for the networks with more than three inputs.

## 2 Problem Formulation

Without loss of generality, we motivate and discuss our approach using the task of learning safe robot controllers for a lower-leg prosthesis. The goal of this task is to learn a policy $\pi_\theta$ which generates control values for the ankle angle of the powered prosthesis given a set of sensor values. Most critically, however, policy $\pi_\theta$ is required to satisfy a set of safety constraints $\Psi$. Fig. 2 provides an overview of our methodology in addressing this challenge. For now, we assume that an unsafe prior policy network may exist, see Fig. 2 (left). The network parameters $\theta$ may be learned with through imitation learning [10], reinforcement learning [28] or any of the common machine learning paradigms. The policy may be optimized for task efficiency, e.g., stable and low-effort walking gaits, but may not yet satisfy any safety constraints. Our goal is to find an adjusted set of network parameters that satisfy any such constraint. We may now choose, for example, to restrict the control rate to specific bounds. Running the network on the input values of a validation data set reveals that violations occur at a number of time steps, as seen in Fig. 2. Our approach, called **NNRepLayer** (Layer-wise Neural Network Repair), takes the original network parameters $\theta$ and the predicates $\Psi$ and yields the updated parameters that generate no violations.

**Notation.** We denote the set of variables $\{a_1, a_2, \cdots, a_N\}$ with $\{a_n\}_{n=1}^N$. Let $\pi_\theta$ be a network policy with $L$ hidden layers. The nodes at each layer $l \in \{l\}_{l=0}^L$ are represented by $x^l$, where $|x^l|$ denotes the dimension of layer $l$ ($x^0$ represents the input of network). The network's output is also denoted by $y$ or $\pi_\theta(x^0)$. We consider fully connected policy networks with weight and bias terms $\{(\theta_w^l, \theta_b^l)\}_{l=1}^{L+1}$. The training data set of $N$ inputs $x_n^0$ and target outputs $t_n$ is denoted by $\{(x_n^0, t_n)\}_{n=1}^N$ sampled from the input-output space $\mathcal{X} \times \mathcal{T} \subseteq \mathbb{R}^{|x^0|} \times \mathbb{R}^{|t|}$. We use $x_n^l$ to denote the vector of nodes at layer $l$ for sample $n$. In this work, we focus on the policy networks with the Rectified Linear Unit (ReLU) activation function $R(z) = \max\{0, z\}$. Thus, given the $n^{\text{th}}$ sample, in the $l^{\text{th}}$ hidden layer, we have $x^l = R\left(\theta_w^l x^{l-1} + \theta_b^l\right)$. An activation function is not applied to the last layer, i.e. $y = \theta_w^{L+1} x^L + \theta_b^{L+1}$.

**Problem Statement (Repair Problem).** Let $\pi_\theta$ be a trained policy network over the training input-output space $\mathcal{X} \times \mathcal{T} \subseteq \mathbb{R}^{|^0|} \times \mathbb{R}^{|t|}$ and $\Psi(y, x^0)$ be a predicate on the output $y$ of network for a set of inputs of interest $x^0 \in \mathcal{X}_r \subseteq \mathcal{X}$. The Repair Problem is to modify the weight $\theta_w$ and bias terms $\theta_b$ of $\pi_\theta$ such that the repaired policy $\pi_{\theta_r}$ satisfies $\Psi(y, x^0)$.

The repair of the policy network should not only satisfy the predicate $\Psi(y, x^0)$ but should also maintain the performance of original policy. To satisfy the latter, the method proposed in [3] ensures the satisfaction of predicate only with a minimal deviation of weights in the last layer. However, as we show latter in the experimental results, the repair of last layer is not necessarily feasible or sufficient to satisfy the predicates with a minimal deviation from the original parameters. Moreover, the minimal deviation from the original weights is not a sufficient guarantee to maintain the original performance of network. It is well-known that subtle changes in the weights may cause the network to significantly deviate from its original performance [29]. Therefore, it is important for the repaired policy $\pi_{\theta_r}$ to also minimize the loss w.r.t. its original training data. We propose NNRepLayer (Layer-wise Neural Network Repair) that satisfies a predicate $\Psi(y, x^0)$ by repairing a specific layer of the policy network while minimizing the training loss.

## 3 NNRepLayer

We can formulate our framework as the minimization problem of the loss function $E(\theta_w, \theta_b)$ subject to $(x^0, t) \in \mathcal{X} \times \mathcal{T}$ and $\Psi(y, x^0)$ for $x^0 \in \mathcal{X}_r$. However, the resulting optimization problem is non-convex and difficult to solve due to the nonlinear ReLU activation function and high-order nonlinear constraints resulted from the multiplication of terms involving the weight/bias variables. In our approach, we obtain a sub-optimal solution by just focusing on repairing a single layer. We therefore modify the weight and bias terms of a single layer to adjust the predictions so as to minimize $E(\theta_w, \theta_b)$ and to satisfy $\Psi(y, x^0)$. Thus, we solve the following problem

**Problem 1.** *Let $\pi_\theta$ denote a trained policy network with $L$ hidden layers over the training input-output space $\mathcal{X} \times \mathcal{T} \subseteq \mathbb{R}^{|x^0|} \times \mathbb{R}^{|t|}$ and $\Psi(y, x^0)$ denote a predicate representing constraints on the output $y$ of $\pi_\theta$ for the set of inputs of interest $x^0 \in \mathcal{X}_r \subseteq \mathcal{X}$. NNRepLayer modifies the weights of a*

*layer $l \in \{1, \cdots, L+1\}$ in $\pi_\theta$ such that the new network $\pi_{\theta_r}$ satisfies $\Psi(y, x^0)$ while minimizing the loss of network $E(\theta_w^l, \theta_b^l)$ with respect to its original training set.*

Since $\mathcal{X}_r$ and $\mathcal{X}$ are not necessarily convex, we formulate NNRepLayer over a data set $\{(x_n^0, t_n)\}_{n=1}^N \sim \mathcal{X} \times \mathcal{T} \cup \mathcal{X}_r \times \tilde{\mathcal{T}}$, where $\tilde{\mathcal{T}}$ is the set of original target values of inputs in $\mathcal{X}_r$. The predicate $\Psi(x^0, y)$ defined over $x^0 \in \mathcal{X}_r$ is not necessarily compatible with the target values in $\tilde{\mathcal{T}}$. It means that the predicate may bound the NN output for $\mathcal{X}_r$ input space such that not allowing an input $x^0 \in \mathcal{X}_r$ to reach its target value in $\tilde{\mathcal{T}}$. It is a natural constraint in many applications. For instance, due to the safety constraints, we may not allow a NN controller to follow its original control reference for a given unsafe set of input states. For a given layer $l$, we also define $E(\theta_w^l, \theta_b^l)$ in the form of sum of square loss $E(\theta_w^l, \theta_b^l) = \sum_{n=1}^N \|y_n(x_n^0, \theta_w^l, \theta_b^l) - t_n\|_2^2$, where $\|\cdot\|_2$ denotes the Euclidean norm. Here, since we only repair the weight and bias terms of target layer $l$, the loss term $E$ is a function of $\theta_w^l$ and $\theta_b^l$, respectively. Hence, the weight and bias terms of all layers except the target layer $l$ are fixed. We define our optimization formulation as follows.

**NNRepLayer Optimization Formulation.** Let $\pi_\theta$ be a neural network with $L$ hidden layers, $\Psi(y, x^0)$ be a predicate, and $\{(x_n^0, t_n)\}_{n=1}^N$ be an input-output data set sampled from $(\mathcal{X} \times \mathcal{T}) \cup (\mathcal{X}_r \times \tilde{\mathcal{T}})$ over the sets $\mathcal{X}$, $\mathcal{X}_r$, $\mathcal{T}$, and $\tilde{\mathcal{T}}$ all as defined in Problem 1. NNRepLayer minimizes the loss (1) by modifying $\theta_w^l$ and $\theta_b^l$ subject to the constraints (2)-(5).

Here, constraint (2) represents the linear forward pass of network's last layer. Constraint (3) represents the forward pass of hidden layers starting from the layer $l$. Except the weight and bias terms of the $l^{\text{th}}$ layer, i.e. $\theta_w^l$ and $\theta_b^l$, the weight and bias terms of the subsequent layers $\{(\theta_w^i, \theta_b^i)\}_{i=l+1}^{L+1}$ are fixed. The sample values of $x_n^{l-1}$ are obtained by the weighted sum of the nodes in its previous layers starting from $x_n^0$ for all $N$ samples $\{n\}_{n=1}^N$. Each ReLU node $x^l$ is formulated using Big-M formulation [30, 31] by

$$(1) \quad \min_{\theta_w^l, \theta_b^l, \delta, y_n, \{x_n^i\}_{i=l}^L, \{\phi_n^i\}_{i=l}^L} E(\theta_w^l, \theta_b^l) + \delta,$$

$$\text{s.t.}$$

$$(2) \quad y_n = \theta_w^{L+1} x_n^L + \theta_b^{L+1},$$

$$(3) \quad x_n^i = R(\theta_w^i x_n^{i-1} + \theta_b^i), \quad \text{for } \{i\}_{i=l}^L$$

$$(4) \quad \Psi(y_n, x_n^0), \quad \text{for } x_n^0 \in \mathcal{X}_r$$

$$(5) \quad \delta \geq \|\theta_w^l - \theta_w^{l,init}\|_\infty, \ \|\theta_b^l - \theta_b^{l,init}\|_\infty \geq 0.$$

$x^l \geq \theta_w^l x_n^{l-1} + \theta_b^l$, $x^l \leq (\theta_w^l x_n^{l-1} + \theta_b^l) - lb(1 - \phi)$, and $x^l \leq ub\,\phi$, where $x^l \in [0, \infty)$, and $\phi \in \{0, 1\}$ determines the activation status of node $x^l$. The bounds $lb, ub \in \mathbb{R}$ are known as Big-M coefficients, $\theta_w^l x_n^{l-1} + \theta_b^l \in [lb, ub]$, that need to be as tight as possible to improve the performance of MIQP solver. We used Interval Arithmetic (IA) Method [32, 14] to obtain tight bounds for ReLU nodes (read the supplementary materials, Sec. A, for further details on IA). Constraint (4) is a given predicate on $y$ defined over $x^0 \in \mathcal{X}_r$. NNReplayer addresses the predicates of the form $\bigvee_{c=1}^C \psi_c(x^0, y)$ where $C$ represents the number of disjunctive propositions and $\psi_i$ is an affine function of $x^0$ and $y$. Finally, constraint (5) bounds the entry-wise max-norm error between the weight and bias terms $\theta_w^l$ and $\theta_b^l$, and the original $\theta_w^{l,init}$ and $\theta_b^{l,init}$ by $\delta$. Considering the quadratic loss function $E(\theta_w^l, \theta_b^l)$ and the affine disjunctive forms of $\Psi(y_n, x_n^0)$ and $R(\theta_w^i x_n^{i-1} + \theta_b^i)$, we solve NNRepLayer as a Mixed Integer Quadratic Program (MIQP).

**Theorem 1.** *Given the predicate $\Psi(y, x^0)$, and the input-output data set $\{(x_n^0, t_n)\}_{n=1}^N$ sampled from $(\mathcal{X} \times \mathcal{T}) \cup (\mathcal{X}_r \times \tilde{\mathcal{T}})$ over the sets $\mathcal{X}$, $\mathcal{X}_r$, $\mathcal{T}$, and $\tilde{\mathcal{T}}$ as defined in Problem 1, assume that $\theta_w^l$ and $\theta_b^l$ are feasible solutions to (1)-(5). Then, $\Psi(\pi_{\theta_r}(x_n^0), x_n^0)$ is satisfied for all input samples $x_n^0$.*

*Proof.* Since the feasible solutions $\theta_w^l$ and $\theta_b^l$ satisfy the hard constraint (4) for the repair data set $\{(x_n^0, t_n)\}_{n=1}^N$, $\Psi(\pi_{\theta_r}(x_n^0), x_n^0)$ is satisfied. $\square$

Given Thm. 1, the following Corollary is straightforward.

**Corollary 1.** *Given the predicate $\Psi(y, x^0)$, and the input-output data set $\{(x_n^0, t_n)\}_{n=1}^N$ sampled from $(\mathcal{X} \times \mathcal{T}) \cup (\mathcal{X}_r \times \tilde{\mathcal{T}})$ over the sets $\mathcal{X}$, $\mathcal{X}_r$, $\mathcal{T}$, and $\tilde{\mathcal{T}}$ as defined in Problem 1, assume that $\theta_w^{l^*}$ and $\theta_b^{l^*}$ are the optimal solutions to the NNRepLayer. Then, for all input samples $x_n^0$ from $\{(x_n^0, t_n)\}_{n=1}^N$, $\Psi(\pi_{\theta_r}(x_n^0), x_n^0)$ is satisfied.*

## 4   Evaluation

We explore the applicability of the framework in satisfying the following three types of constraints. **Global constraints** that encode global bounds on the network's output, i.e., $y \in [y_{min}, y_{max}]$. **Input-output constraints** that ensure the network's output $y$ to stay within a certain bound with respect to the network's input $x^0$, i.e., $\{\psi_c(x^0, y) \leq 0\}_{c=0}^C$, where $C$ is the number of constraints and $\psi_c$ is an affine function of $x^0$ and $y$. Finally, **conditional constraints** that encode if-then-else constraints described as $\{\psi_c(x^0, y) \leq 0, \text{ if } x^0, y \in S_c\}_{c=0}^C$, where $C$ specifies the number of conditions, $\psi_c$ is an affine function of $x^0$ and $y$, and $S_c \subseteq \mathcal{X} \times \mathcal{T}$. We designed a number of experiments to validate that our repair framework can successfully apply these constraints to the policy network. Following our motivation, all experiments were performed on the prosthetic walking gait generation task introduced in Fig. 2. Through these experiments we aim to answer the following questions: (1) Does our method enable the Prosthetic device to address all the three types of aforementioned constraints? (2) Can the repaired controller be employed in a real walking scenario successfully? (3) How robust is the policy repaired through our technique against the unseen constraint-violating samples?

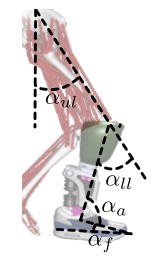

Figure 3: Prosthetic device model.

**Experimental Setup.**   In our experiment, we train a policy network $\pi_\theta$ for controlling a prosthesis, which then undergoes the repair process to ensure compliance with the safety constraints. To this end, we first train the model using an imitation learning [33] strategy. For data collection, we conducted a study approved by the Institutional Review Board (IRB), in which we recorded the walking gait of a healthy subject without any prosthesis. Walking data included three inertial measurement units (IMUs) mounted via straps to the upper leg (Femur), lower leg (Shin), and foot. The IMUs acquired both the angle and angular velocity of each limb portion in the world coordinate frame at 100Hz. Ankle angle $\alpha_a$ was calculated as a post process from the foot and lower limb IMUs. We then trained the NN to generate the ankle angle from upper and lower limb IMU sensor values. More specifically, the NN model receives the angle and velocity from the upper and lower limb sensors (network inputs $x^0$), $\alpha_{ul}, \dot{\alpha}_{ul}, \alpha_{ll}, \dot{\alpha}_{ll}$, respectively, to predict the ankle angle $\alpha_a$ (network output $y$) which is, later, used as the control parameter for a PD controller on the prosthetic. See Fig. 3 for a visualization of the individual sensor readings. We used a sliding window of input variables, denoted as $dt$ ($dt = 10$ in all our experiments), to account for the temporal influence on the control parameter and to accommodate for noise in the sensor readings. Therefore, the input to the network is $dt \times |x^0|$, or more specifically the current and previous $dt$ sensor readings. In all experiments, we trained a three-hidden-layer deep policy network with 32 ReLU nodes at each hidden layer. After the networks were fully trained we assessed the policy for constraint violations and collected samples for NNRepLayer. We tested NNRepLayer on the last and the second to the last layer of network policy to satisfy the constraints with a subset of the original training data including both adversarial and non-adversarial samples. In all experiments, we used 150 samples in NNRepLayer and a held out set of size 2000 for testing. Finally, the repaired policies to satisfy global and input-output constraints are tested on a prosthetic device for 10 minutes of walking, see Fig. 5. More specifically, the same healthy subject was fitted with an ankle bypass; a carbon fiber structure molded to the lower limb and constructed such that a prosthetic ankle can be attached to allow the able-bodied subject to walk on the prosthesis, as shown in Fig. 2. The extra weight and off-axis positioning of the device incline the individual towards slower, asymmetrical gaits that generates strides out of the original training distribution [34, 10]. The participant is then asked to walk again for 10 minutes to assess whether constraints are satisfied.

To evaluate if the repair can be generalized to the unseen adversarial samples, we analyze the *violation degree*. The violation degree is measured as the distance of the network output with respect to the constraint set. For each experiment, we explain how this distance between the output and the constraint set is calculated. We compared our framework with retraining, fine-tuning [26, 35, 27, 7], and the patch-based repair method in [4] (REASSURE). Adversarial samples in the repair data set are hand-labeled for fine-tuning and retraining so that the target outputs satisfy the given predicates. In fine-tuning, as proposed in [26, 35], we used the collected adversarial data set to train all the parameters of the original policy by gradient descent using a small learning rate ($10^{-4}$). To avoid over-fitting to the adversarial data set, we trained the weights of the top layer first, and thereafter fine-tuned the remaining layers for a few epochs. The same hand-labeling

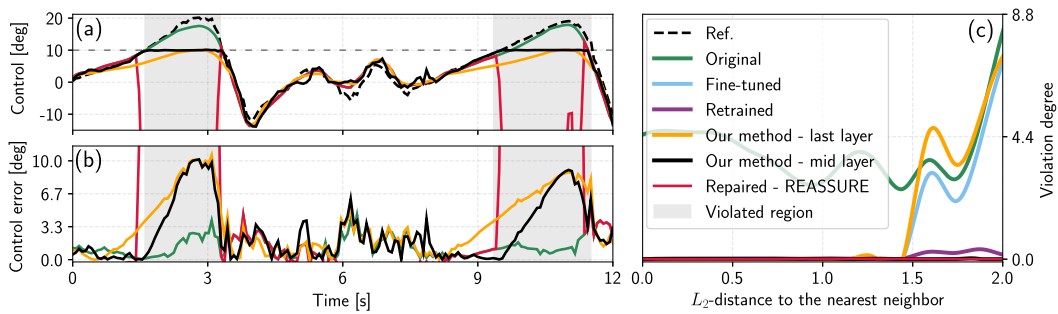

Figure 4: Global constraint: (a) ankle angle, $\alpha_a$, (b) the error between the predicted and the reference controls, (c) the violation degree vs. $L_2$-distance between the test and repair sample inputs.

strategy is applied in retraining, except that a new policy is trained from scratch for all original training samples. In both methods, we trained the policy until all the adversarial samples in the repair data set satisfy the given predicates on the network's output. Our code is available on GitHub: https://github.com/k1majd/NNRepLayer.git.

### 4.1 Experiments and Results

**Global Constraint.** The global constraint ensures that the prosthesis control, i.e., $\alpha_a$, stays within a certain range and never outputs an unexpected large value that disturbs the user's walking balance. Additionally, the prosthetic device we utilized in these scenarios contains a parallel compliant mechanism. As such, either the human subject or the robotic controller could potentially drive the mechanism into the hard limits, potentially damaging the device. In our walking tests, we therefore specified global constraints such that the ankle angle stays within the bounds of $[-14, 24]$ [deg] regardless of whether it is driven by the human or the robot. In simulation experiments, we enforced artificially strict bounds on the ankle angle $\alpha_a$ to never exceed $\alpha_a = 10$ [deg] which is a harder bound to satisfy. We defined the degree of violation as 0 if $\alpha_a \in [\alpha_a^{min}, \alpha_a^{max}]$, and $\min\{|\alpha_a - \alpha_a^{max}|, |\alpha_a - \alpha_a^{min}|\}$, otherwise. As shown in Fig. 4 (a)-(b), the repaired network successfully satisfies the constraints in the original faulty regions while maintaining the tracking performance of the controller in the unconstrained regions. Figure 4 (c) demonstrates that the violation degree stays almost zero for even distant originally violating data points after repairing the mid layer. The red control signal in Fig. 5 also shows that our method successfully imposes the control bounds $[-14, 24]$ to the actual prosthesis walking test.

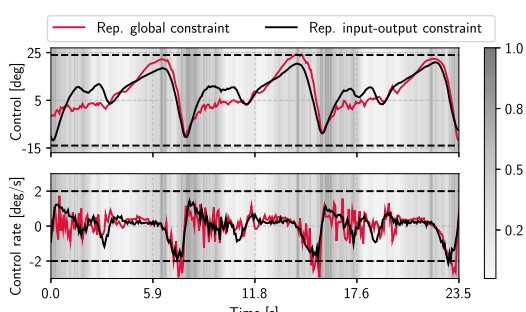

Figure 5: Real prosthesis walking test results for imposing the global constraint of $[-14, 24]$ to the control (shown in red) and bounding the control rate by 2 [deg/s] (shown in black). The color bar represents the normalized $L_2$-distance of each test input to its nearest neighbor in the repair set.

**Input-output Constraint.** Deep neural networks as highly non-linear function approximators have the ability to change the outputs more rapidly than what is feasible for the robotic prosthesis or for the human subject to accommodate. Therefore, we propose an additional constraint over the possible change of control actions from one time-step to the next. This constraint should act to both smooth the control action in the presence of sensor noise, as well as to reduce hard peaks and oscillations in the control action. To capture this constraint as an input-output relationship, we trained the policy network by adding the previous $dt$ control actions $\{\alpha_a(i)\}_{i=t-dt}^{t-1}$ as inputs to the policy network along the values of upper and lower limb sensors. Imposed constraints in this example follow the form $|\alpha_a(t) - \alpha_a(t-1)| \leq \Delta\alpha_a^{max}$. In prosthetic walking tests, we bounded the control rate by $\Delta\alpha_a^{max} = 2$ [deg/s], and in our simulations we tested $\Delta\alpha_a^{max} = 1.5$ and $\Delta\alpha_a^{max} = 2$ [deg/s]. Our

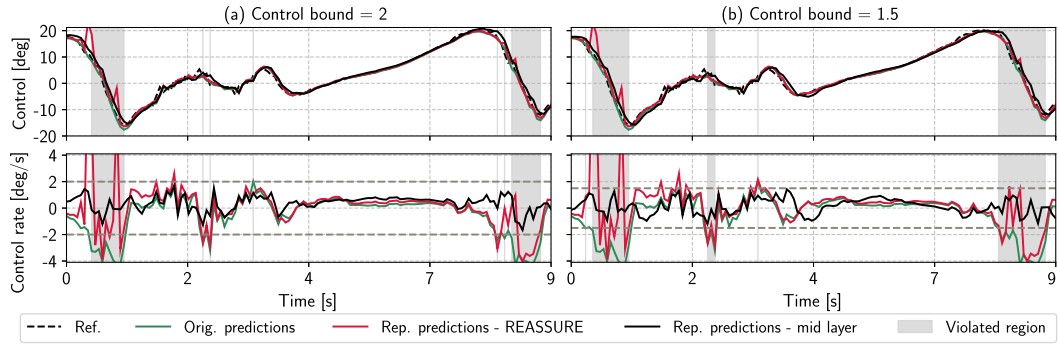

Figure 6: Input-output constraint: Ankle angles and Ankle angle rates for bounds (a) $\Delta\alpha_a = 2$ and (b) $\Delta\alpha_a = 1.5$

simulation results in Fig. 6 demonstrate that NNRepLayer satisfies both bounds on the control rate which subsequently results in a smoother control output. It can also be observed that NNRepLayer successfully preserves the tracking performance of controller. In this experiment, we defined the violation degree as 0 if the $|\Delta\alpha_a(t)| \leq \Delta\alpha_a^{max}$ and $|\Delta\alpha_a(t) - \Delta\alpha_a^{max}|$, otherwise. Figure 7 (a) demonstrates that the violation degree of NNRepLayer is almost zero by increasing the distance of the test samples from the repair set. Same results are obtained in the actual prosthetic walking tests as shown in Fig. 5. The satisfaction of input-output constraints (black signal) is achieved even for the samples that are distant from the repair set. Finally, in this experiment, applying NNRepLayer to the last layer does not obtain even a feasible solution to the optimization problem (1)-(5).

**Conditional Constraint.** Depending on the ergonomic needs and medical history of a patient, the attending orthopedic doctor or prosthetist may identify certain body configurations that are harmful, e.g., they may increase the risk of osteoarthritis or musculoskeletal conditions [36, 37]. Following this rationale, we define a region $\mathcal{S}$ of joint angles space that should be avoided. An example of such a region is demonstrated in Fig. 8 as a grey box $\mathcal{S} = \{(\alpha_{ul}, \alpha_a) \mid \alpha_{ul} \in [-2, -0.5], \alpha_a \in [1, 3]\}$ in the joint space of ankle and femur angles. To satisfy this constraint the control rate should be tuned such that the joint ankle and femur angles stay out of set $\mathcal{S}$. This constraint can be defined as an if-then-else proposition $\alpha_{ul} \in [-2, -0.5] \implies (\alpha_a \in (-\infty, 1]) \vee (\alpha_a \in [3, \infty))$ which can be formulated as the disjunction of linear inequalities on the network's output. For each given test input and its corresponding output $\alpha_a$, the degree of violation is defined as the distance of $\alpha_a$ to the box if $\alpha_a$ is outside the box, and 0 otherwise. Figure 8 demonstrates the output of new policy after repairing with NNRepLayer. As it is shown, our method avoids the joint ankle and femur angles to enter the unsafe region $\mathcal{S}$. Figure 7 (b) also illustrates low output violation degree for the distant test input samples from the repair input set. Finally, we observed that repairing the last layer does not result in a feasible solution.

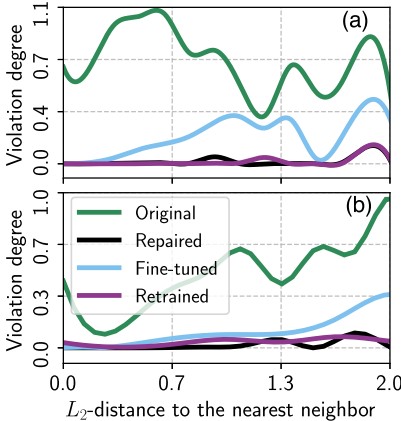

Figure 7: The violation degree vs. $L_2$-distance between the test and repair sample inputs for (a) input-output constraint, and (b) conditional constraint cases.

**Comparison w\Fine-tuning, Retraining, and REASSURE.** In each experiment, we demonstrated the violation degree and the control signals of our method compared with fine-tuning, retraining [26, 35, 27, 7], and REASSURE [4]. Comparing to [4], while REASSURE guarantees the satisfaction of constraints in the local repaired linear regions, we showed that this method significantly reduces the performance of network in the repaired local regions, see Figures 4 and 6. This method cannot address the input-output constraints given the faulty samples, and it introduces 500 times more faulty samples compared to our technique. REASSURE cannot also accommodate the conditional constraints. Unlike REASSURE that guarantees the

Table 1: The table reports: RT: runtime, MAE: Mean Absolute Error between the repaired and the original outputs, RE: the percentage of adversarial samples that are repaired (Repair Efficacy), and IB: the percentage of test samples that were originally safe but became faulty after the repair (Introduced Bugs). The metrics are the average of 50 runs.

| | NNRepLayer | | | | REASSURE [4] | | | |
| | RT [s] | MAE | RE [%] | IB [%] | RT [s] | MAE | RE [%] | IB [%] |
|---|---|---|---|---|---|---|---|---|
| Global | $233 \pm 159$ | $1.4 \pm 0.11$ | $99 \pm 1$ | $0.09 \pm 0.20$ | $14 \pm 1$ | $2.3 \pm 0.78$ | $97 \pm 1$ | $0$ |
| Input-output | $112 \pm 122$ | $0.5 \pm 0.03$ | $98 \pm 1$ | $0.19 \pm 0.18$ | $30 \pm 8$ | $0.6 \pm 0.03$ | $19 \pm 4$ | $85 \pm 5$ |
| Conditional | $480 \pm 110$ | $0.35 \pm 0.07$ | $93 \pm 2$ | $0.11 \pm 0.26$ | Infeasible | Infeasible | Infeasible | Infeasible |

| | Fine-tune | | | | Retrain | | | |
| | RT [s] | MAE | RE [%] | IB [%] | RT [s] | MAE | RE [%] | IB [%] |
|---|---|---|---|---|---|---|---|---|
| Global | $25 \pm 13$ | $1.2 \pm 0.03$ | $97 \pm 4$ | $0.95 \pm 0.45$ | $127 \pm 30$ | $1.4 \pm 0.08$ | $98 \pm 3$ | $0.65 \pm 0.40$ |
| Input-output | $8 \pm 2$ | $0.6 \pm 0.03$ | $88 \pm 2$ | $2.47 \pm 0.49$ | $101 \pm 1$ | $0.5 \pm 0.04$ | $98 \pm 1$ | $0.28 \pm 0.32$ |
| Conditional | $18 \pm 3$ | $0.7 \pm 0.10$ | $72 \pm 5$ | $0.27 \pm 0.25$ | $180 \pm 2$ | $0.31 \pm 0.03$ | $76 \pm 2$ | $0.12 \pm 0.35$ |

satisfaction of constraints for the samples in the same linear region as the repaired samples, our technique only guarantees the satisfaction of constraints for the repaired samples. While we empirically showed the generalizability of our technique in a local neighborhood of the repaired samples, our method does not theoretically guarantee the satisfaction of constraints for the unseen adversarial samples. We proposed a sound algorithm in the supplementary materials, Sec. B, that guarantees the safety for all other unseen samples. Table 1 better illustrates the success of our method in satisfying the constraints while maintaining the control performance.

As shown in Table 1, retraining and NNRepLayer both perform well in maintaining the minimum absolute error and the generalization of constraint satisfaction to the unseen testing samples for global and input-output constraints. However, the satisfaction of if-then-else constraints is challenging for retraining and fine-tuning as the repair efficacy is dropped by almost 30% using these techniques. It also highlights the power of our technique in generalizing the satisfaction of conditional constraints to the unseen cases. For further details on the comparison results, read the supplementary materials, Sec. C.

## 5   Conclusion & Discussion

In this paper, we introduced an algorithm for training safe neural network controllers that satisfy a formal set of safety constraints. Our approach, NNRepLayer, performs a global optimization step in order to perform layer-wise repair of neural

Figure 8: Enforcing the conditional constraints to keep the joint femur-ankle angles out of the grey box.

network weights. In real-robot experiments, we have shown that the introduced methodology produces safe neural policies for a lower-leg prosthesis satisfying a variety of constraints. We argue that this type of approach is critical for human-centric and safety-critical applications of robot learning, e.g., the next-generation of assistive robotics.

**Discussion.** The introduced approach does not generally ensure that for *any* input the constraints will be satisfied. Instead it guarantees this property for all data points provided at the time of repair. Hence, proper care has to be taken to ensure that the repair process involves representative samples of the variety of inputs seen in the application domain. From a computational vantage point, solving the MIQP underlying NNRepLayer is a demanding process which scales with the size of the network. In our experiments, we successfully repaired NN layers with up to 256 neurons, with global optimization taking between multiple minutes and up to 10 hours (read the supplementary materials, Sec. D, for the detailed experimental results). Moreover, we showed in the supplementary materials, Sec. E, that the repair of randomly selected sub-nodes of a hidden layer can accurately repair the network in much shorter time (more that 12 times faster than the full repair). Finally, our approach is limited to repairing individual layers in a network. Early results on iteratively repairing multiple layers are promising and will be reported in the future. However, our approach cannot simultaneously repair multiple layers or the entire network.

**Acknowledgments**

This work was partially supported by the National Science Foundation under grants CNS-1932068, IIS-1749783, and CNS-1932189.

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
