# OpenReview forum: "Safe Robot Learning in Assistive Devices through Neural Network Repair"
_robot-learning.org/CoRL/2022/Conference — CoRL 2022 Poster_

### Official Review · Reviewer_4TJw · 2022-07-12

**Originality:** Good
**Technical Quality:** Very Good
**Clarity Of Presentation:** Good
**Impact:** 4

**Recommendation:**

Weak Accept: I recommend accepting the paper, but will not argue for my recommendation if the majority of other reviewers have a different opinion.

**Summary:**

This paper presents safe-learning-based controllers for a prosthesis device. It proposed NNReplLayer (Layer-wise Neural Network Repair) that modifies the weights and biases of the neural network to avoid violations.
This study performed experiments with a real prosthesis and a healthy subject under three different constraints.
I also compared the proposed method with the fine-tuned and retrained ones.

**Issues:**

As described in the weakness, please provide enough surveys for the safe learning control (or comparisons with the other methods).
Besides, please clarify the impacts of the results by adding explanations and plots to demonstrate the benefits of safe prosthesis control.

**Quality Of The Limitations Section:**

Limitations are addressed clearly

**Reviewer Expertise:**

3: The reviewer is fairly confident that the evaluation is correct

**Robotics Focus:**

Sufficient demonstration on hardware

**Strengths And Weaknesses:**

STRENGTH
This paper is well written and tackles significant problems for prosthesis devices. The limitations are clearly presented. As this paper describes, few studies consider the safety of learning-based control for prosthesis devices. Meanwhile, it proposes an exciting idea by employing recent network repair techniques.

WEAKNESS
Although this study presents an attractive idea, the impacts of the results look weak. As suggested in the following lists, more explanations or evaluations are required for acceptance.

a) My main concern is whether this method is the best for a safe assistive device controller. Recent studies have demonstrated many safe-RL or safe imitation learning approaches (e.g., the review paper might be helpful [r1], but not limited to this paper). Although few cases use safe learning for assistive devices, I wonder if this study's setting can adopt other safe learning techniques.
However, this study lacks discussions about safe learning approaches. Please survey more related works and describe whether the network repair is the best way. Besides, it will be much better to compare the existing safe-learning controllers.

[r1] Brunke, L., Greeff, M., Hall, A. W., Yuan, Z., Zhou, S., Panerati, J., & Schoellig, A. P. (2021). Safe learning in robotics: From learning-based control to safe reinforcement learning. arXiv preprint arXiv:2108.06266. https://arxiv.org/pdf/2108.06266.pdf

b) The proposed method successfully regulates the control inputs; however, the difference between the proposed and retrained network does not seem significant as the retrained network also showed a small violation degree in global and input-output constraints.
This paper needs more explanations or results to describe the significance. For example, how about plotting the other joints or trunk? I would like to see whether the user shows unstable walking when a violation occurs.
Besides, it would be better to plot the control and control rate in the conditional constraints like Figs. 4 and 7.

c) The horizontal axes of figures 4, 5, and 7 show the time [s]. I suspect they should be timestep as the plot shows a few strides. The user might walk incredibly slowly if the scale is time [s].

d) The term of the title: "assistive devices," would have a broader meaning than this study's scope. I would recommend using the "prosthesis device" instead.

e) The proposed section is well and generally described; however, it would be better to explain what the variable (e.g., x, y, and t) stands for in the prosthesis context. The readers will be easier to follow.

f) Figs. 4 and 7 are difficult to see the differences in each condition. Please make them larger.

g) In Fig. 4 (c), the repaired last layer in global constraints showed a higher violation degree. Please provide the reason or discussions for the negative performance.


**Summary Of Recommendation:**

This paper is well written, and the core idea is interesting. However, I am concerned about the impact of the results, the relationship to other safe learning, and the benefits in the prosthesis contexts, leading to weak reject recommendations. More explanations and/or comparisons might increase the grade after the rebuttal.

After rebuttal comment

Thank the authors for addressing my comment. All of my concerns have been solved. I changed my recommendation to weak acceptance.

---

> ### Author Response · Authors · 2022-08-21
> **Responses to reviewer 4TJw**
>
> We thank the reviewer for the insightful and positive feedbacks.
> We are encouraged that the reviewer found our paper and idea to be clearly presented, well-written,
> tackled a significant problem.
> We are pleased that the reviewer recognized the importance of the problem
> that we addressed and the originality of our repair technique.
>
> We addressed the reviewer comments below and will incorporate all feedbacks.
>
> 1. **Our method vs safe RL:**
> We thank the reviewr for introducing the reference [r1] that is a good summary/survey of current approaches to safe learning.
> Within this research area one can distinguish two main different research directions:
> (i) learning for control (reinforcement learning, learning based model predictive control, etc), and
> (ii) learning of safety certificates. Our work is conceptually different from both these research problems.
> At a very conceptual level, in learning for control approaches (model based or model free),
> the agents can control their actions and explore the state space. The goal is to learn the control inputs/actions
> that will optimize some cost or reward function and will not violate the safety constraints.
> In our setting (i.e. imitation learning) the data is already provided to us and we do not have an explicit cost or reward function.
> Since the cost/reward function is not available, we must learn a policy immediately from the provided samples
> and have no access to the environment.
> Our imitation learning approach ensures that safety constraints will be imposed on top of the learned behavior
> while making sure that the demonstrated behavior (the training data) is only minimally modified.
> Learning safety certificates/invariants is also not within our scope of work
> since we know that the network already violates some of the constraints that we need to impose.
> We argue that safety in RL (reference [r1]) and safety in imitation learning (our paper) are complementary approaches.
> In RL, every interaction with the environment, i.e. rollout, exposes the agent to risks (potential injuries to the human wearing the prosthetic).
> Hence, it is beneficial to start with a controller that guarantees absence of violations on an initial set of training data.
> As a result, the RL agent is exposed to to substantially fewer risks
> and may only need a small number of iterations
> to ensure safety on newly encountered adversarial examples in the domain of application.
>
> 2. **The horizontal axes of figures 4, 5, and 7 show the time [s]. I suspect they should be timestep as the plot shows a few strides:**
>  Yes, the horizontal axes labels show timestep. Thanks for your comment. We will fix it in the final version.
>
> 3. **In Fig. 4 (c), the repaired last layer in global constraints showed a higher violation degree.
> Please provide the reason or discussions for the negative performance:**
> The last layer only has an scaling effect, thereby, results in smaller error reduction compared to repairing the hidden layers.
>
> **References**
>
> [r1] Brunke, L., Greeff, M., Hall, A. W., Yuan, Z., Zhou, S., Panerati, J., & Schoellig, A. P. (2021). Safe learning in robotics: From learning-based control to safe reinforcement learning. arXiv preprint arXiv:2108.06266. https://arxiv.org/pdf/2108.06266.pdf

---

> > ### Comment · Reviewer_4TJw · 2022-08-26
> > **Response to Authors**
> >
> > Thank the authors for sincerely addressing my comments. My main concern has been solved.
> > However, I still have a question about my comment b). How did this study collect adversarial samples for the fine-tuning and retaining approaches in the walking experiments? Did it provide any safety for this?
> > If this adversarial data collection is dangerous and careful operations are required, network repair is very useful. Otherwise, the fine-tuning and retaining approaches are simple and useful enough.
> > I would like to know the detail of the procedures for the adversarial data collection.

---

> > > ### Author Response · Authors · 2022-08-27
> > > **Authors' response to the follow up question by reviewer 4TJw**
> > >
> > > We are glad the main concern of the reviewer is addressed.
> > >
> > > For this specific application, we impose hard constraints over training data collected through experiments. In other words, the training data are samples of the process we would like to replicate, but in addition we want to impose additional safety constraints. Hence, we can search over the training data and see which violate our constraints before the NN is utilized in a control loop. With our method, these adversarial samples are used for NN repair before the NN is deployed. At the end, we have the guarantee that the network is at least repaired for these samples, and, moreover, we have established experimentally that our repair generalizes better than retraining and fine-tuning.
> > >
> > > As a side note, it is important to stress that retraining and fine-tuning sometimes cannot repair the NN even over the provided adversarial samples and that it is not easy to generate safe training samples from the adversarial samples. That is, since imposing hard constraints using fine-tuning and retraining is not easy, we hand-labeled the output references of the adversarial input samples to stay within the constraint. Hand-labeling should be performed carefully to ensure the learned mapping maintains the performance of original controller [r1, r2]. However, there is no straightforward mechanism to do so.
> > >
> > > At the end, safety could be verified by running a verification algorithm on the NN such as [r3 and r4] to ensure the generalizability of our method. For further detailed discussion we refer to our response to the reviewer TbCe, item 3 and the supplementary materials Section 4.
> > >
> > > [r1] A. Sinitsin, V. Plokhotnyuk, D. Pyrkin, S. Popov, and A. Babenko. Editable neural networks. arXiv preprint arXiv:2004.00345, 2020.
> > >
> > > [r2] X. Ren, B. Yu, H. Qi, F. Juefei-Xu, Z. Li, W. Xue, L. Ma, and J. Zhao. Few-shot guided mix for dnn repairing. In 2020 IEEE International Conference on Software Maintenance and Evolution (ICSME), pages 717–721. IEEE, 2020.
> > >
> > > [r3] Katz, Guy, et al. "Reluplex: An efficient SMT solver for verifying deep neural networks." International conference on computer aided verification. Springer, Cham, 2017.
> > >
> > > [r4] Huang, Xiaowei, et al. "Safety verification of deep neural networks." International conference on computer aided verification. Springer, Cham, 2017.

---

> > > > ### Author Response · Authors · 2022-08-27
> > > > **Final comment**
> > > >
> > > > We thank the reviewer again for the thoughtful comments. We hope that our detailed responses warrant a revision of the review scores.

---

### Official Review · Reviewer_qM6j · 2022-07-29

**Originality:** Very Good
**Technical Quality:** Very Good
**Clarity Of Presentation:** Excellent
**Impact:** 4

**Recommendation:**

Weak Accept: I recommend accepting the paper, but will not argue for my recommendation if the majority of other reviewers have a different opinion.

**Summary:**

The authors present a method for updating/repairing a neural network such that the outputs on given examples do not violate safety constraints.  They show that the method is sound and that it produces safer trajectories in a combination of simulated and real-world experiments with prostheses.

**Issues:**

Answer the following questions:

How is the layer that is repaired chosen?
Can the computational expense be justified a little more clearly?
How might the results be the same (or different) with an actual prosthesis user?
Discuss a little more how having representative samples would result in a generally safe controller? Can you be sure that if two examples are close (in some sense) and one of them is repaired, that the second would be repaired too?

**Quality Of The Limitations Section:**

Limitations are addressed clearly

**Reviewer Expertise:**

3: The reviewer is fairly confident that the evaluation is correct

**Robotics Focus:**

Sufficient demonstration on hardware

**Strengths And Weaknesses:**

Strengths
- Tested on an actual prosthesis
- Nice mix of empirical and theoretical results
- This approach seems like it could generalize to other domains as well and may be of interest to robot learning in general


Weaknesses
- The method doesn't actually produce guarantees on unseen examples; it only guarantees that known violations have been repaired.  This leaves it a bit up in the air whether the method would actually result in safe NN controllers in practice
- I understand that it is normal to evaluate such algorithms with nondisabled participants, but I would at least like to see some discussion on how the results might be different with an actual prosthesis user (e.g., how would this work with someone with whom you can't collect "correct" gait data because they are a double lower-limb amputee or because they don't have a "normal" gait)
- the method is very computationally expensive

Minor notes
- Fig 4 subfigures are not labeled

**Summary Of Recommendation:**

The paper makes an interesting contribution to deep learning for robotics; although expensive, the method is shown to improve over simply re-training a network especially for complex constraints (which often show up in robotics domains).  The approach is validated with simulated and empirical results although not with an actual prosthesis user.

---

> ### Author Response · Authors · 2022-08-21
> **Responses to reviewer qM6j - part 1**
>
> ## Main Concerns:
>
> We are glad that the reviewer found our method generalizable to other domains,
> implemented on an actual real-world robot control application,
> and empowered by a mix of empirical and theoretical guarantees.
> We are pleased that the reviewer recognized the originality,
> technical quality, and the clarity of this paper.
>
> We addressed the reviewer comments below and will incorporate all feedbacks.
>
> 1. **Generalization to the unseen samples:**
> Yes, our technique only guarantees the satisfaction of constraints for
> the repaired samples which we believe is still a significant contribution.
> Additionally, to guarantee the generalization of our method, NNRepLayer
> can be employed in a loop with a sound verifier [r1 and r2] such that our method first returns the repaired network.
> Then, the verifier evaluates the network.
> If the algorithm terminates, the network is guaranteed to be safe for all other unseen samples in the target input space.
> Otherwise, the network is not satisfied to be safe and the verifier provides the newly found adversarial samples
> for which the guarantees do no hold.
> In turn, NNRepLayer uses the given samples by the verifier to repair the network.
> This loop terminates when the verifier confirms the satisfaction of constraints.
> This algorithm guarantees the soundness of our technique.
> _**For further detailed discussion we refer to our response to the reviewer TbCe, item 3 and the supplementary materials Section 4.**_
>
> 2. **How the results might be different with an actual prosthesis user:**
> The generalization of repaired neural networks to unseen, especially out of distribution samples,
> is of great importance. To ensure generalization we both trained and repaired our control networks
> over a wide array of gait parameters by instructing the subjects during data collection
> to walk at varying speeds and stride lengths.
> When tested, we utilized a prosthetic bypass (Figure 1 in the paper),
> and though it was on an able-bodied individual the added weight ensures
> that the resulting strides are out of the original training distribution.
> Our results, therefore, show that our NN repair method generalizes well to unseen out of distribution samples,
> and specifically to asymmetric gaits.
> _**We will include these details in the “Experimental Setup” subsection in the Sec. 4 of the final version.**_
>
> 3. **Computational complexity of our technique:**
> Yes, our method is more computationally demanding than finetuning and retraining since these latter methods do not take hard constraints into account
> However, as we mentioned in the paper (Sec. 5),
> we successfully repaired a network with up to 64 neurons that took up to 6 hours.
> To better demonstrate the scalability of our method,
> we conducted another experiment with 1000 repair samples on a
> network with 256 nodes in each hidden layer to satisfy the input-output constraint
> (we terminate the solver after 10 hours and report the best found feasible solution).
> Our experimental results demonstrate that our technique repaired a network with up to 256 nodes with 100% repair efficacy in 10 hours.
> Similar network structure and sizes are frequently used in robotics and controls tasks
> for example researchers in Google Brain trained a robot locomotion task with a network of size 256,
> Haarnoja et al, 2019 [r3] (2 hidden layer, 256 nodes).
> Other examples include
> Fernandez et al, 2020 [r4] (3 hidden layer, 256 nodes),
> Landgraf et al, 2021 [r5] (2 hidden layer, 64 nodes),
> Pinosky et al [r6] (2 hidden layer, 200 nodes),
> Zimmer et al [r7] (2 hidden layer, 50 nodes),
> and Kristoffersen et al [r8] (2 hidden layer, 50 nodes).  _**Our detailed experimental results on the network with 256 hidden nodes
> is presented in the supplementary materials, section 1.**_

---

> > ### Author Response · Authors · 2022-08-21
> > **Responses to reviewer qM6j - part 2**
> >
> > ## Issues:
> >
> > 4. **How is the layer that is repaired chosen?**
> >
> > 	**a)** We are currently investigating methods to detect and repair only a specified sub-nodes or sub-weights
> > 	of the target layer instead of repairing the full layer.
> > 	It results in tighter MIQP formulations and can drastically reduce the number of integer variables.
> > 	This is the scope of our future work but to demonstrate the effectiveness of this heuristic,
> > 	we repaired 10 randomly selected nodes in a network with 64 hidden nodes for 35 times.
> > 	We terminated the solver after 30 minutes and reported the results.
> > 	To detect the sparse nodes that can satisfy the constraints,
> > 	we solved the original full repair by adding the taxicab error norm ($l_1$ norm) of repaired weights
> > 	with respect to their original values to the MIQP cost function.
> > 	We showed that repairing the obtained sparse nodes reached a cost value very close
> > 	to the cost value of the originally full repair problem,
> > 	**in only 30 minutes** versus 6 hours.
> > 	Our experiment also shows that the repair of randomly selected nodes can accurately repair the network in much shorter time
> > 	(30 mins versus 6 hours) while reaching an even higher repair efficacy with respect to the full repair.
> > 	_**These results are presented in the supplementary materials, section 3.**_
> >
> > 	**b)** In our future work, we aim to explore the neural network pruning techniques
> > 	(that were originally employed to reduce the size of network while maintaining the network’s accuracy)
> > 	[r9, and r10] to measure the contribution of each neuron to the network’s misbehavior
> > 	and obtain just a small effective sub-node of the network to repair.
> > 	We believe this method can not only detect which layer to repair
> > 	but it also can identify the nodes that can satisfy the faulty behavior.
> >
> > **References**
> >
> > [r1] Katz, Guy, et al. "Reluplex: An efficient SMT solver for verifying deep neural networks." International conference on computer aided verification. Springer, Cham, 2017.
> >
> > [r2] Huang, Xiaowei, et al. "Safety verification of deep neural networks." International conference on computer aided verification. Springer, Cham, 2017.
> >
> > [r3] Haarnoja, Tuomas, et al. "Learning to Walk Via Deep Reinforcement Learning." Robotics: Science and Systems. (2019).
> >
> > [r4] Fernandez, Gabriel I., et al. "Deep Reinforcement Learning with Linear Quadratic Regulator Regions." arXiv preprint arXiv:2002.09820 (2020).
> >
> > [r5] Landgraf, Christian, et al. "A reinforcement learning approach to view planning for automated inspection tasks." Sensors 21.6 (2021): 2030.
> >
> > [r6] Pinosky, Allison, et al. "Hybrid control for combining model-based and model-free reinforcement learning." The International Journal of Robotics Research (2022): 02783649221083331.
> >
> > [r7] Zimmer, Matthieu, Yann Boniface, and Alain Dutech. "Developmental reinforcement learning through sensorimotor space enlargement." 2018 Joint IEEE 8th International Conference on Development and Learning and Epigenetic Robotics (ICDL-EpiRob). IEEE, (2018).
> >
> > [r8] Kristoffersen, Morten B., et al. "User training for machine learning controlled upper limb prostheses: a serious game approach." Journal of NeuroEngineering and Rehabilitation 18.1 (2021): 1-15.
> >
> > [r9] LeCun, Yann, John Denker, and Sara Solla. "Optimal brain damage." Advances in neural information processing systems 2 (1989).
> >
> > [r10] Hassibi, Babak, David Stork, and Gregory Wolff. "Optimal brain surgeon: Extensions and performance comparisons." Advances in neural information processing systems 6 (1993).

---

> > > ### Author Response · Authors · 2022-08-27
> > > **Final comment**
> > >
> > > We thank the reviewer again for the thoughtful comments. We hope that our detailed responses warrant a revision of the review scores.

---

> > ### Comment · Reviewer_qM6j · 2022-08-25
> > **Response to Authors**
> >
> > Thank you to the authors for their response.  I think this is probably not the most compelling argument that this would work in the real world (a nondisabled person walking faster or slower is pretty different from a disabled person walking), but I hope the authors can show their work having a significant benefit over other approaches in the target domain soon.

---

> > > ### Author Response · Authors · 2022-08-26
> > > **Response to reviewer qM6j regarding the benefits of using prosthetic bypass**
> > >
> > > Thank you for your comment. The utilization of a prosthetic bypass by an able-bodied individual does in fact mimic amputee walking in several ways. Numerous state-of-the-art prosthesis research papers have utilized prosthetic bypasses in prosthesis control research [r1 and r2]. Actually, prosthetic bypass is not only **a widely used device in simulating amputee walking** but it is also **employed in the testing and validation of the designed prosthesis devices** [r3, r4, r5, and r6]. The extra weight and off-axis positioning of the device incline the individual towards slower, asymmetrical gaits. Additionally, the removal of force feedback from the subject’s foot suitably imitates the conditional effects of an amputee. Since this is still early research our goal is not to say that our controllers work best on all amputees, but rather to show quantitatively that our method works on neural network controllers for assistive devices and is capable of enforcing constraints on said neural network controllers. Lots of work and many papers are ahead to push this research to amputee populations and also to look into the myriad of edge cases that it brings. We thank the reviewer again for the thoughtful comments. We hope our responses and the cited papers have convinced the reviewer of the significance of our results.
> > >
> > > [r1] Cortino, R. J., Bolívar-Nieto, E., Best, T. K., & Gregg, R. D. (2022, May). Stair Ascent Phase-Variable Control of a Powered Knee-Ankle Prosthesis. In 2022 International Conference on Robotics and Automation (ICRA) (pp. 5673-5678). IEEE.
> > >
> > > [r2] Gao, C., Gehlhar, R., Ames, A. D., Liu, S. C., & Delbruck, T. (2020, May). Recurrent neural network control of a hybrid dynamical transfemoral prosthesis with EdgeDRNN accelerator. In 2020 IEEE International Conference on Robotics and Automation (ICRA) (pp. 5460-5466). IEEE.
> > >
> > > [r3] Grimmer, M., Holgate, M., Holgate, R., Boehler, A., Ward, J., Hollander, K., ... & Seyfarth, A. (2016). A powered prosthetic ankle joint for walking and running. Biomedical engineering online, 15(3), 37-52.
> > >
> > > [r4] Cempini, M., Hargrove, L. J., & Lenzi, T. (2017, September). Design, development, and bench-top testing of a powered polycentric ankle prosthesis. In 2017 IEEE/RSJ International Conference on Intelligent Robots and Systems (IROS) (pp. 1064-1069). IEEE.
> > >
> > > [r5] Lenzi, T., Sensinger, J., Lipsey, J., Hargrove, L., & Kuiken, T. (2015, August). Design and preliminary testing of the RIC hybrid knee prosthesis. In 2015 37th Annual International Conference of the IEEE Engineering in Medicine and Biology Society (EMBC) (pp. 1683-1686). IEEE.
> > >
> > > [r6] Upadhye, S., Shah, C., Liu, M., Buckner, G., & Huang, H. H. (2021, September). A Powered Prosthetic Ankle Designed for Task Variability–A Concept Validation. In 2021 IEEE/RSJ International Conference on Intelligent Robots and Systems (IROS) (pp. 6153-6158). IEEE.

---

### Official Review · Reviewer_pwYF · 2022-07-31

**Originality:** Fair
**Technical Quality:** Fair
**Clarity Of Presentation:** Good
**Impact:** 2

**Recommendation:**

Strong Reject: I recommend rejecting the paper and will argue for my recommendation even if other reviewers hold a different opinion.

**Summary:**

This paper proposes an algorithm to repair neural networks to ensure they satisfy certain properties.  Given a trained neural network that violates some specification, the paper uses mixed-integer programming to change the weights of some layers to ensure that the resulting network satisfies the specification. The paper provides some numeric examples using a prosthesis walking assistive robot to support their proposed framework.

**Issues:**

- Is the MIQP problem always feasible? How to choose which layer to repair? What happens if the NN can not be repaired by adjusting the weights of a single layer?

- While the numerical analysis tries to check the rate of the adversarial examples before and after the repair, several critical pieces of information are not present. In particular, the experiments should report the fraction of adversarial examples introduced by the repair, i.e., how many non-adversarial inputs became adversarial after the repair. This can be done by evaluating the original NN on the adversarial inputs found in the repaired one to check whether these adversarial examples were introduced by the repair.


**Quality Of The Limitations Section:**

Additional details required

**Reviewer Expertise:**

5: The reviewer is absolutely certain that the evaluation is correct and very familiar with the relevant literature

**Robotics Focus:**

Relevant but unlikely to deploy to hardware in near future

**Strengths And Weaknesses:**

Strengths:
- Applying the repair algorithm to a real-world example like prosthesis walking is interesting.

Weaknesses:
- Compared to the literature, the proposed method provides weak guarantees. Other algorithms in the literature (which the paper cited only 3 of them) ensure the resulting NN is correct *for all inputs* while the proposed method provides such a guarantee to *the samples available in the dataset*. There is no guarantee that the proposed method will not introduce more counterexamples in areas where the original NN (before the repair) was correct.

- The paper provides no comparison against the other NN repair techniques.

- The MIQP formulation is straightforward and has been extensively used in the literature for NN verification and repair. While other papers in the literature try to provide novel heuristics to better find the solution to these MIQP problems, this paper only applies off-the-shelf solvers.

- The complexity of the proposed MIQP program grows with the number of samples in the dataset. This seems to be why the framework was applied to simplistic settings (150 samples and 64 neurons). Other techniques in the literature do not depend on the dataset size and hence do not suffer from such complexity.

**Summary Of Recommendation:**

- The novelty of the proposed framework is limited.
- The theoretical guarantee of the proposed framework is weaker than those in the literature.
- Lack of comparison with the results in the literature.
- Algorithmic complexity

---

> ### Author Response · Authors · 2022-08-21
> **Responses to reviewer pwYF - part 1**
>
> We are glad that the reviewer found the real-world application of our technique to be interesting.
>
> We addressed the reviewer's comments below and will incorporate all feedbacks.
>
> ## Main Concerns:
>
> 1. **Compared to the literature, the proposed method provides weak guarantees.
> Other algorithms in the literature (which the paper cited only 3 of them)
> ensure the resulting NN is correct for all inputs
> while the proposed method provides such a guarantee to the samples available in the dataset.**
> To the best of our knowledge,
> there exist no repair method in the literature
> that guarantees the satisfaction of constraints for all inputs other than the samples available in the dataset
>  or in a local neighborhood of the faulty samples. There are only two repair methods that provide local soundness guarantees:
>
> 	**a) Fu et al. [r1]:** this method guarantees the constraint satisfaction for the other faulty inputs
> 	that lie in the same linear regions as the repaired samples.
> 	However, another unseen faulty behavior that does not belong to any of the repaired linear regions is not repaired.
> 	_**We compared NNRepLayer with this technique and the results are available in the supplementary materials Section 5.**_
> 	As illustrated, this method performs poorly in satisfying constraints for the samples that are not used in repair.
> 	This method also introduces 500 times more faulty samples compared to our technique for input-output constraints.
>
> 	**b) Sotoudeh et al. [r2]**: this paper proposed a decoupled DNN architecture that decouples the activation of neurons from their values.
> 	It repairs linear regions related to each faulty sample similar as [r1].
> 	Therefore, this method also poorly generalizes to the unrepaired faulty samples.
> 	Besides, this method does not isolate the repair of faulty region from the rest of input space.
> 	Therefore, it only guarantees the satisfaction of constraints in the faulty linear region
> 	but may introduce more counterexamples in other linear regions.
> 	Additionally, and most importantly, this method causes the repaired network to be discontinuous.
> 	Therefore, it is not applicable in robot learning and control applications which is the application target of this paper.
> 	Running the code provided for [r2] on our experiments produced errors since, as the authors mentioned in [r2],
> 	this method is only applicable for either large networks with one-dimensional input or mid-sized networks
> 	with two-dimensional inputs (our networks include >40 inputs).
> 	_**For the detailed comparison, please read supplementary materials, Section 5.**_
>
> 	**c)** We also asked the codes for [r3] through email since the link they provided in the paper does not exist.
> 	We have not still received any response from the authors.
>
> 	**d)** Finally, for generalization guarantees of our technique.
> 	_**We presented an algorithm in the supplementary materials, Section 4, that guarantees the soundness of our technique.**_
> 	This algorithm employs NNRepLayer in a loop with a sound verifier [r4 and r5] such that our method first returns the repaired network.
> 	Then, the verifier evaluates the network.
> 	If the algorithm terminates, the network is guaranteed to be safe for all other unseen samples in the target input space.
> 	Otherwise, the network is not satisfied to be safe and the verifier provides the newly found adversarial samples
> 	for which the guarantees do no hold.
> 	In turn, NNRepLayer uses the given samples by the verifier to repair the network.
> 	This loop terminates when the verifier confirms the satisfaction of constraints. This algorithm guarantees the soundness of our technique.
> 	Unfortunately, we did not have enough time to emplement the algorithm on our actual setup
> 	in order to report the statistical results as well.
> 	_**We refer to our response to reviewer TbCe, item 3,
> 	and our newly added algorithm in the supplementary materials Section 4 for the further details.**_
>
> **Note:** We searched again through the literature,
> and we acknowledge that we missed citing [r2] and [r3].
> We will cite these papers in our final revision.
> _**We are happy to compare our method with other references that the reviewer may provide.**_

---

> > ### Author Response · Authors · 2022-08-21
> > **Responses to reviewer pwYF - part 2**
> >
> > 2. **Comparison against the other NN repair techniques:**
> > _**The detailed comparison of our technique with other repair techniques is presented in the supplementary materials Section 5.**_
> > The Table 3 in Section 5 of the supplementary materials shows that our method is the only repair method in the literature
> > that can impose complicated hard constraints to the network.
> > This makes NNRepLayer applicable in the robot learning and control applications.
> > We also ran the repair codes from REASSURE [r1] and PRDNN [r2] on our problem.
> > Overall, the repair methods [r1 and r2] are either infeasible [r2] to be applied to the network sizes we used
> > (only applicable to the networks with one or two dimensional inputs as authors mentioned in [r2])
> > or perform poorly such that they cannot accommodate the complicated constraints that we address [r1].
> > We contacted the authors of [r1] to ensure the accuracy of formulated repair using their technique.
> > We also asked the codes for [r3] through email since the link they provided in the paper does not exist.
> > We have not still received any response from the authors.
> > _**We are happy to compare our method with other references that the reviewer may provide.**_
> > _**Note that all communications are documented and can be presented to the reviewer upon request.**_
> >
> > 3. **MIQP formulation and other heuristics to better find the solution:**
> > Unlike the verification techniques that employ MIQP to search the input space of network,
> >  we derived MIQP in our peorblem to search the weights space of network.
> >  To the best of our knowledge no other method in the literature has used MIQP to formulate the repair problem **in the weight space**.
> >  _**please read our more detailed response in item 6.**_
> >
> > 4. **Other techniques in the literature do not depend on the dataset size and hence do not suffer from such complexity:**
> > We kindly ask the reviewer to elaborate more on this comment.
> > If the reviewer means computational complexity, the computational complexity of all repair techniques grows
> > at least linearly with respect to the dataset size.
> > Therefore, they all depend on the dataset size.
> > Also, in our latest experiments we successfully repaired a network with 256 hidden neurons and 1000 repair samples
> > that shows the significance of our results and the application of our technique in many robot learning and robot control tasks
> > which is the target of this paper.
> > _**For the latest results please read the supplementary materials Section 1 and our response to the reviewer TbCe item 1.**_
> >
> > 5. **The reviewer’s score for Robotics Focus - Relevant but unlikely to deploy to hardware in near future:**
> > We respectfully disagree with this score. Note that while our technique generalizes to the other safety-critical applications,
> > the focus of this paper is on the applications of neural network repair in assistive devices.
> > We already showed through extensive experimental results that this method has been deployed to a real prosthesis device
> > (Figure 5 in the paper). Similar network structure and sizes are frequently used in robotics and
> > control tasks for example researchers in Google Brain trained a robot locomotion task with a network of size 256
> > , Haarnoja et al, 2019 [r7] (2 hidden layer, 256 nodes).
> > Other examples include Fernandez et al, 2020 [r8] (3 hidden layer, 256 nodes),
> > Landgraf et al, 2021 [r9] (2 hidden layer, 64 nodes),
> > Pinosky et al [r10] (2 hidden layer, 200 nodes),
> > Zimmer et al [r11] (2 hidden layer, 50 nodes),
> > and Kristoffersen et al [r12] (2 hidden layer, 50 nodes).

---

> > > ### Author Response · Authors · 2022-08-21
> > > **Responses to reviewer pwYF - part 3**
> > >
> > > 6. **The novelty of the proposed framework is limited:**
> > > We respectfully disagree with this evaluation for the following reasons:
> > >
> > > 	**a)** To the best of our knowledge, this is the first paper that
> > > 	employs neural network repair in real robot control and robot learning applications
> > > 	(We are happy to compare our method with other references that the reviewer may provide).
> > > 	We actually showed the usefulness and the value of neural network repair in real control tasks while other methods
> > > 	only reported the results of running their technique on standard baselines.
> > > 	While the application of our technique is currently limited to the networks with up to 256 neurons,
> > > 	it still can be employed in a wide variety of control tasks.
> > > 	We refer to our response in item 5.
> > > 	Note that the largest network that is repaired by the methods
> > > 	that guarantee the satisfaction of constraints for the detected faulty
> > > 	samples has 256 hidden nodes in each layer. Therefore,
> > > 	we showed that we can repair networks with at least
> > > 	the same size as the largest network in the literature [r1, r2, r3, r13, r14].
> > >
> > > 	**b)** As the reviewer mentioned, MIP is a standard technique in verification.
> > > 	All the problems in verification, including reachability and robustness,
> > > 	explore the input space of the network.
> > > 	However, our method employs the MIP formulation to formulate repair
> > > 	so as to search the weight space of network.
> > > 	This way we empower the neural network to ensure the satisfaction of detected faulty samples
> > > 	by applying hard constraints to the output of network.
> > > 	To the best of our knowledge, no repair technique in the literature has used MIQP to formulate the repair problem **in the weight space**
> > > 	(We are happy to compare our method with other references that the reviewer may provide).
> > > 	_**We also proposed some techniques (heuristics) to improve the solution time that
> > > 	we discussed in detail in our response to the reviewer TbCe, item 2.**_
> > >
> > > ## Issues
> > >
> > > 7. **How to choose which layer to repair?**
> > >
> > > 	**a)** We are currently investigating methods to detect and repair only a specified sub-nodes or sub-weights
> > > 	of the target layer instead of repairing the full layer.
> > > 	It results in tighter MIQP formulations and can drastically reduce the number of integer variables.
> > > 	This is the scope of our future work but to demonstrate the effectiveness of this heuristic,
> > > 	we repaired 10 randomly selected nodes in a network with 64 hidden nodes for 35 times.
> > > 	We terminated the solver after 30 minutes and reported the results.
> > > 	To detect the sparse nodes that can satisfy the constraints,
> > > 	we solved the original full repair by adding the taxicab error norm ($l_1$ norm) of repaired weights
> > > 	with respect to their original values to the MIQP cost function.
> > > 	We showed that repairing the obtained sparse nodes reached a cost value very close
> > > 	to the cost value of the originally full repair problem,
> > > 	**in only 30 minutes** versus 6 hours.
> > > 	Our experiment also shows that the repair of randomly selected nodes can accurately repair the network in much shorter time
> > > 	(30 mins versus 6 hours) while reaching an even higher repair efficacy with respect to the full repair.
> > > 	_**These results are presented in the supplementary materials, section 3.**_
> > >
> > > 	**b)** In our future work, we aim to explore the neural network pruning techniques
> > > 	(that were originally employed to reduce the size of network while maintaining the network’s accuracy)
> > > 	[r15, and r16] to measure the contribution of each neuron to the network’s misbehavior
> > > 	and obtain just a small effective sub-node of the network to repair.
> > > 	We believe this method can not only detect which layer to repair
> > > 	but it also can identify the nodes that can satisfy the faulty behavior.
> > > 8. **The experiments should report the fraction of adversarial examples introduced by the repair:**
> > > We evaluated the “introduced bugs” by our method that measures the percentage of non-adversarial inputs
> > > that became adversarial versus the number of test samples. _**The new results are presented in Table 2,
> > > Section 5 of the supplementary materials.**_
> > > As demonstrated in the Table, the testing evaluation of the repair method [r1]
> > > (that guarantees the local generalization) introduces 500 times more faulty samples compared
> > > to our technique for the input-output constraints.

---

> > > > ### Author Response · Authors · 2022-08-21
> > > > **Responses to reviewer pwYF - part 4**
> > > >
> > > > **References**
> > > >
> > > > [r1] Fu, Feisi, and Wenchao Li. "Sound and Complete Neural Network Repair with Minimality and Locality Guarantees." International Conference on Learning Representations. 2021.
> > > >
> > > > [r2] Sotoudeh, Matthew, and Aditya V. Thakur. "Provable repair of deep neural networks." Proceedings of the 42nd ACM SIGPLAN International Conference on Programming Language Design and Implementation. 2021.
> > > >
> > > > [r3] Usman, Muhammad, et al. "NN repair: Constraint-Based Repair of Neural Network Classifiers." International Conference on Computer Aided Verification. Springer, Cham, 2021.
> > > >
> > > > [r5] Katz, Guy, et al. "Reluplex: An efficient SMT solver for verifying deep neural networks." International conference on computer aided verification. Springer, Cham, 2017.
> > > >
> > > > [r6] Huang, Xiaowei, et al. "Safety verification of deep neural networks." International conference on computer aided verification. Springer, Cham, 2017.
> > > >
> > > > [r7] Haarnoja, Tuomas, et al. "Learning to Walk Via Deep Reinforcement Learning." Robotics: Science and Systems. (2019).
> > > >
> > > > [r8] Fernandez, Gabriel I., et al. "Deep Reinforcement Learning with Linear Quadratic Regulator Regions." arXiv preprint arXiv:2002.09820 (2020).
> > > >
> > > > [r9] Landgraf, Christian, et al. "A reinforcement learning approach to view planning for automated inspection tasks." Sensors 21.6 (2021): 2030.
> > > >
> > > > [r10] Pinosky, Allison, et al. "Hybrid control for combining model-based and model-free reinforcement learning." The International Journal of Robotics Research (2022): 02783649221083331.
> > > >
> > > > [r11] Zimmer, Matthieu, Yann Boniface, and Alain Dutech. "Developmental reinforcement learning through sensorimotor space enlargement." 2018 Joint IEEE 8th International Conference on Development and Learning and Epigenetic Robotics (ICDL-EpiRob). IEEE, (2018).
> > > >
> > > > [r12] Kristoffersen, Morten B., et al. "User training for machine learning controlled upper limb prostheses: a serious game approach." Journal of NeuroEngineering and Rehabilitation 18.1 (2021): 1-15.
> > > >
> > > > [r13] Dong, Guoliang, et al. "Towards repairing neural networks correctly." 2021 IEEE 21st International Conference on Software Quality, Reliability and Security (QRS). IEEE, (2021).
> > > >
> > > > [r14] Goldberger, Ben, et al. "Minimal Modifications of Deep Neural Networks using Verification." LPAR. Vol. 2020. (2020).
> > > >
> > > > [r15] LeCun, Yann, John Denker, and Sara Solla. "Optimal brain damage." Advances in neural information processing systems 2 (1989).
> > > >
> > > > [r16] Hassibi, Babak, David Stork, and Gregory Wolff. "Optimal brain surgeon: Extensions and performance comparisons." Advances in neural information processing systems 6 (1993).

---

> > > > > ### Author Response · Authors · 2022-08-27
> > > > > **Final comment**
> > > > >
> > > > > We thank the reviewer again for the comments. We hope that our detailed responses warrant a revision of the review scores.

---

> > ### Comment · Reviewer_pwYF · 2022-08-27
> > **Reply to the comparison against other techniques**
> >
> > I would like to thank the authors for the additional results. However, these results do not speak to the concern I raised. While other work in the literature tries to provide theoretical guarantees on the repair beyond the samples (for example, repairing the local neighborhood of the faulty samples or all the ones within a linear predicate), this paper provides none. It only focuses on repairing the ones exactly at the samples and not even in their neighborhood which *theoretically* is a weaker guarantee than the ones in the literature.
> >
> > For item (1.d), I can not comment on this new algorithm since the authors did not provide any experimental results that shows the feasibility of this algorithm.

---

> > > ### Author Response · Authors · 2022-08-27
> > > **We respectfully disagree with this evaluation**
> > >
> > > We respectfully disagree with this characterization!
> > >
> > > There are only two repair techniques in the literature that guarantee the generalizability of repair over a local linear region of the **found** faulty samples [r1 and r2].
> > >
> > > Sotoudeh et al [r1], as the paper authors mentioned, is not even applicable to be applied to the networks with more than two inputs!
> > >
> > > We compared our technique with Fu et al [r2] empirically and showed that: while this technique guarantees the satisfaction of constraints in the local repaired linear regions, we showed that this method significantly reduces the performance of network in the repaired local regions (Supplementary materials, Fig. 3, Section 5)! We also showed that this method performs poorly in generalizing the repair for the input-output and conditional constraints.
> > >
> > > Consequently, if the basis of this review is on the empirical evaluation, we already showed that our method outperforms the techniques in the literature with local guarantee. But if the review is based on theoretical guarantees, we proved that Algorithm 1 (and Theorem 1) in the supplementary materials verifies the safety of network for all other unseen samples.
> > >
> > > [r1] Sotoudeh, Matthew, and Aditya V. Thakur. "Provable repair of deep neural networks." Proceedings of the 42nd ACM SIGPLAN International Conference on Programming Language Design and Implementation. 2021.
> > >
> > > [r2] Fu, Feisi, and Wenchao Li. "Sound and Complete Neural Network Repair with Minimality and Locality Guarantees." International Conference on Learning Representations. 2021.

---

### Official Review · Reviewer_TbCe · 2022-08-01

**Originality:** Good
**Technical Quality:** Good
**Clarity Of Presentation:** Very Good
**Impact:** 2

**Recommendation:**

Weak Accept: I recommend accepting the paper, but will not argue for my recommendation if the majority of other reviewers have a different opinion.

**Summary:**

In order to force neural networks to satisfy certain constraints in safety-critical applications, the authors propose directly modifying weights so that satisfaction of the constraints is formally guaranteed. They achieve this goal by formulating the repair problem as a Mixed Integer Quadratic Program problem and finding an analytic solution. They find that their method outperforms fine-tuning on adherence to a number of constraints, and outperforms retraining the network from scratch on adherence to conditional constraints. They validate this result on a real prosthetic leg.

**Issues:**

I understand that given a neural network to repair, NNRepLayer always returns the same set of weights. However, it still depends on the pre-trained network, which is randomly initiallized and whose training is partially stochastic. As such, the experiments really should be repeated across multiple trained networks to measure statistical significance.

It's also a little unclear what some of the baselines are. I assume "retrain"  means "retrain a new randomly-initialized network from scratch", but this is not made explicit and leaves some room for ambiguity.

**Quality Of The Limitations Section:**

Limitations are addressed clearly

**Reviewer Expertise:**

3: The reviewer is fairly confident that the evaluation is correct

**Robotics Focus:**

Sufficient demonstration on hardware

**Strengths And Weaknesses:**

The paper introduces an interesting alternative to fine-tuning, which involves using Mixed Integer Quadratic Programming to enforce constraints. I haven't seen any approaches before attempting to solve modify neural nets using MIQP, but the application to safety-critical systems makes a great deal of sense. The main appeal of this method seems to be that it is possible to guarantee that if an answer is returned, then the constraint will be exactly satisfied at all known data points, rather than approximately satisfied, as would be the case for fine-tuning or retraining. This is definitely an appealing quality.

This just seems like an absolutely crushingly long time to run on a width 32 MLP, such that it's really hard to imagine this approach scaling up to networks of any significant size. Furthermore, because this is solved by an NP complete method, it doesn't seem like there are many ways that clever optimizations could reduce the runtime. I would find this tradeoff more acceptable if the solver was use to produce a guarantee that the neural network can never produce outputs that violate the constraint for all inputs. After all, this is for systems that prioritize safety, so it makes sense to trade off training time for theoretical guarantees. But this method still has to account for generalization, because the guarantee only applies for the training set.

**Summary Of Recommendation:**

The work is interesting, and hopefully offers future avenues into using NNs for safety critical applications. At present the method does not seem usable for any but the smallest networks, and offers only narrow improvements over the presumably-faster method of retraining a network from scratch. However, this niche of small safety-critical networks does have its uses (such as the prosthetic leg mentioned here). For this reason, I recommend acceptance.

---

> ### Author Response · Authors · 2022-08-21
> **Responses to reviewer TbCe - part 1**
>
> We thank the reviewer for the insightful feedback.
> We are happy that the reviewer found our work interesting and novel.
> We are glad that the reviewer recognized the importance of our repair technique and found it insightful
> and essential for guaranteeing the safety of DNNs in safety critical applications.
>
> We addressed the reviewer comments below and will incorporate all feedbacks.
>
> ## Main Concerns:
>
> 1. **Computational complexity of our technique:**
> Yes, our method is more computationally demanding than finetuning and retraining since these latter methods do not take hard constraints into account.
> However, as we mentioned in the paper (Sec. 5),
> we successfully repaired a network with up to 64 neurons that took up to 6 hours.
> To better demonstrate the scalability of our method,
> we conducted another experiment with 1000 repair samples on a
> network with 256 nodes in each hidden layer to satisfy the input-output constraint
> (we terminate the solver after 10 hours and report the best found feasible solution).
> Our experimental results demonstrate that our technique repaired a network with up to 256 nodes with 100% repair efficacy in 10 hours.
> Similar network structure and sizes are frequently used in robotics and controls tasks
> for example researchers in Google Brain trained a robot locomotion task with a network of size 256,
> Haarnoja et al, 2019 [r1] (2 hidden layer, 256 nodes).
> Other examples include
> Fernandez et al, 2020 [r2] (3 hidden layer, 256 nodes),
> Landgraf et al, 2021 [r3] (2 hidden layer, 64 nodes),
> Pinosky et al [r4] (2 hidden layer, 200 nodes),
> Zimmer et al [r5] (2 hidden layer, 50 nodes),
> and Kristoffersen et al [r6] (2 hidden layer, 50 nodes).  _**Our detailed experimental results on the network with 256 hidden nodes
> is presented in the supplementary materials, section 1.**_
>
> 2. **It doesn't seem like there are many ways that clever optimizations could reduce the runtime:**
> No, there are different techniques that can decrease the computational time of repair solver:
>
>      **a)** We can terminate the MIQP program early and still obtain a good feasible solution with small MIP gap
> 	 (the difference between the lower and upper objective bounds in branch-and-bound MIP solving method).
> 	 Also,
> 	 finding tight bounds for the decision variables can improve the solution time.
> 	 In our method,
> 	 we used interval arithmetic for finding tight bounds on each neuron for each sample by bounding the change of weights by $l_{\infty}$ norm.
> 	 _**For further details on interval arithmetic method, please read the supplementary materials, section 2.**_
>
> 	 **b)** We can only repair specified sub-nodes or sub-weights of the target layer instead of repairing the full layer.
> 	 It results in tighter MIQP formulations and can drastically reduce the number of integer variables.
> 	 This is the scope of our future work but to demonstrate the effectiveness of this heuristic,
> 	 we repaired 10 randomly selected nodes in a network with 64 hidden nodes for 35 times.
> 	 We terminated the solver after 30 minutes and reported the results.
> 	 To detect the sparse nodes that can satisfy the constraints,
> 	 we solved the original full repair by adding the taxicab error norm ($l_1$ norm)
> 	 of repaired weights with respect to their original values to the MIQP cost function.
> 	 We showed that repairing the obtained sparse nodes reached a cost value very close to the cost value of the originally full repair problem,
> 	 **in only 30 minutes** versus 6 hours.
> 	 Our experiment also shows that the repair of randomly selected nodes can accurately repair the network in much shorter time
> 	 (30 mins versus 6 hours) while reaching an even higher repair efficacy with respect to the full repair.
> 	 _**These results are presented in the supplementary materials, section 3.**_
>
> 	 **c)** In our future work, we aim to explore the neural network pruning techniques
> 	 (that were originally employed to reduce the size of network while maintaining the network’s accuracy)
> 	 [r7, and r8] to measure the contribution of each neuron to the network’s misbehavior and obtain just
> 	 a small effective sub-node of the network to repair.
> 	 We believe this method can not only detect which layer to repair but it also can identify the nodes that can satisfy the faulty behavior.

---

> > ### Author Response · Authors · 2022-08-21
> > **Responses to reviewer TbCe - part 2**
> >
> > 3. **Generalization to the unseen samples:**
> > Yes, our technique only guarantees the satisfaction of constraints for the repaired samples
> > which we believe is still a significant contribution.
> > To our best of knowledge no other repair technique in the literature also guarantees the generalization
> > to the unseen samples _**(for the detailed comparison of our technique with other repair methods, we refer to the supplementary materials
> > Section 5)**_.
> > Moreover, our empirical simulation results show that our repair technique generalizes well
> > to the unseen samples (paper, Table 1),
> > the highest repair efficacy compared to Fine-tuning and retraining.
> > Besides, the hand-labeling mechanism in retraining and fine-tuning cannot accurately characterize the constraints
> > (as we showed these methods fail to capture the conditional constraints)
> > while our method explicitly formulates the hard constraints.
> >
> > Finally, to guarantee the generalization of our method, NNRepLayer can be employed in a loop with a sound verifier
> > [r9 and r10] such that our method first returns the repaired network.
> > Then, the verifier evaluates the network.
> > If the algorithm terminates, the network is guaranteed to be safe for all other unseen samples in the target input space.
> > Otherwise, the network is not satisfied to be safe and the verifier provides the newly found adversarial samples
> > for which the guarantees do no hold.
> > In turn, NNRepLayer uses the given samples by the verifier to repair the network.
> > This loop terminates when the verifier confirms the satisfaction of constraints.
> > This algorithm guarantees the soundness of our technique.
> > _**We present this algorithm in the supplementary materials Section 4.**_
> >
> > ## Issues:
> >
> > 4. **The experiments should be repeated across multiple trained networks to measure statistical significance:**
> > Thanks, we applied your suggestion. **Please find the updated results in the Table 2 of the supplementary materials.**
> >
> > 5. **It's also a little unclear what some of the baselines are.
> > I assume "retrain" means "retrain a new randomly-initialized network from scratch",
> > but this is not made explicit and leaves some room for ambiguity:**
> > Yes, in our experiments we retrained a new randomly initialized network from scratch using the original training samples and hand-labeling.
> > This highlights another drawback of retraining since it requires all the training samples to repair the faulty behavior
> > (18200 samples in our experiments).
> > However, we just modify the originally trained network with a quite smaller number of samples (150 samples).
> >
> > **References**
> >
> > [r1] Haarnoja, Tuomas, et al. "Learning to Walk Via Deep Reinforcement Learning." Robotics: Science and Systems. (2019).
> >
> > [r2] Fernandez, Gabriel I., et al. "Deep Reinforcement Learning with Linear Quadratic Regulator Regions." arXiv preprint arXiv:2002.09820 (2020).
> >
> > [r3] Landgraf, Christian, et al. "A reinforcement learning approach to view planning for automated inspection tasks." Sensors 21.6 (2021): 2030.
> >
> > [r4] Pinosky, Allison, et al. "Hybrid control for combining model-based and model-free reinforcement learning." The International Journal of Robotics Research (2022): 02783649221083331.
> >
> > [r5] Zimmer, Matthieu, Yann Boniface, and Alain Dutech. "Developmental reinforcement learning through sensorimotor space enlargement." 2018 Joint IEEE 8th International Conference on Development and Learning and Epigenetic Robotics (ICDL-EpiRob). IEEE, (2018).
> >
> > [r6] Kristoffersen, Morten B., et al. "User training for machine learning controlled upper limb prostheses: a serious game approach." Journal of NeuroEngineering and Rehabilitation 18.1 (2021): 1-15.
> >
> > [r7] LeCun, Yann, John Denker, and Sara Solla. "Optimal brain damage." Advances in neural information processing systems 2 (1989).
> >
> > [r8] Hassibi, Babak, David Stork, and Gregory Wolff. "Optimal brain surgeon: Extensions and performance comparisons." Advances in neural information processing systems 6 (1993).
> >
> > [r9] Katz, Guy, et al. "Reluplex: An efficient SMT solver for verifying deep neural networks." International conference on computer aided verification. Springer, Cham, 2017.
> >
> > [r10] Huang, Xiaowei, et al. "Safety verification of deep neural networks." International conference on computer aided verification. Springer, Cham, 2017.

---

> > > ### Author Response · Authors · 2022-08-27
> > > **Final comment**
> > >
> > > We thank the reviewer again for the thoughtful comments. We hope that our detailed responses warrant a revision of the review scores.

---

### Meta-Review · Area_Chair_UXFr · 2022-08-15

**Recommendation:** Accept (Poster)
**Confidence:** 4

**Metareview:**

The reviewers are leaning towards accepting this paper. I appreciate the thorough responses from the authors. I also appreciate the supplementary material which made the paper stronger.

The reviewer pwYF is still against this work. However, I believe that the authors' response is convincing (could be better if it is more polite).

Summary: The reviewer pwYF has a strong opinion against this work. The reviewer made many valid points and the authors should answer them clearly. In general, the reviewers have many common concerns regarding the theoretical guarantees and the computation cost.

Quality: On average, the reviewers rated this paper fair. Some reviewers argued that this paper needs to show more evidence to back up the claims

Clarity: All reviewers rated that this paper is clear.

Originality: The reviewer pwYF mentioned that MIQP has been used for NN verification and repair extensively. The authors should address this comment.

Significance: Overall, the reviewers questioned the impact of this work. Many reviewers questioned the significance or usefulness of the safety guarantee. The method is safe only on the training data and there is no guarantee that it will be safe on the real hardware. The reviewers are concerned about the computation cost of the method. They generally agree that it is not going to be scalable to bigger data sizes.  The reviewer 4TJw mentioned that there are more suitable approaches for safe assistive device control with a reference.

**Best Paper Nomination:**

No

---

> ### Author Response · Authors · 2022-08-27
> **Response to area chair**
>
> We would like to thank the area chair for the time and effort spent on collecting the reviews and summarizing them.
>
> We believe that the harsh criticism by reviewer pwYf on originality, importance, and robotics focus is not merited. Here, we provide a summary of our responses to the main concerns raised by reviewer pwYf:
>
> **1. MIQP has been used for NN verification and repair:** First, repair and verification are two different problems. In general, verification explores the input space of network to obtain tight bounds over the output of network. However, repair searches over the NN parameters to ensure the satisfaction of constraints by the network’s output. Second, despite our extensive initial literature review, we searched again to find repair methods using MIQP following the reviewer’s comments. However, no other repair technique in the literature has used MIQP to formulate the repair problem. As we mentioned in our response to reviewer pwYF (item 6.b), we are happy to compare our method with other references that the reviewer may provide.
>
> **2. Novelty:** Besides being the first paper that formulates repair with MIQP, to the best of our knowledge, this is the first paper that employs neural network repair in real robot control and robot learning applications. As we mentioned in our response to reviewer pwYF (item 6.a), we are happy to compare our method with other references that the reviewer may provide.
>
> **3. The reviewer’s score for Robotics Focus - Relevant but unlikely to deploy to hardware in near future:** We already showed through extensive experimental results that this method has been deployed to a real prosthesis device (Figure 5 in the paper). Similar network structure and sizes are frequently used in robotics and control tasks for example researchers in Google Brain trained a robot locomotion task with a network of size 256, Haarnoja et al, 2019 [r1] (2 hidden layer, 256 nodes). Other examples are referenced in our response to reviewer pwYF, item 5.
>
> **4. Comparison against other repair techniques:** The detailed comparison of our technique versus the other repair techniques in the literature is provided in the supplementary materials Section 5. We showed that our method is the only repair method in the literature that can impose complicated hard constraints to the network. There are only two repair techniques in the literature that guarantee the generalizability of repair over a local linear region of found faulty samples [r2 and r3]. Sotoudeh et al [r2], as the paper authors mentioned, is not even applicable to be applied to the networks with more than two inputs. We compared our technique with Fu et al [r3] empirically and showed that this method significantly reduces the performance of network in the repaired local regions (Supplementary materials, Fig. 3, Section 5). We also showed that this method performs poorly in generalizing the repair for the input-output and conditional constraints.  Therefore, if the basis of this review is on the empirical evaluation, we already showed that our method outperforms the techniques in the literature with local guarantee. But if the review is based on theoretical guarantees, we proved that Algorithm 1 (and Theorem 1) in the supplementary materials verifies the safety of network for all other unseen samples.
>
>
> Summary of our responses to the common concerns:
>
> **5. Computational complexity:** Yes, our method is more computationally demanding than finetuning and retraining since these latter methods do not take hard constraints into account. To better demonstrate the scalability of our method, we conducted another experiment with 1000 repair samples on a network with 256 nodes in each hidden layer to satisfy the input-output constraint (detailed in the supplementary materials Section 1). As we mentioned in item 3, Similar network structure and sizes are frequently used in robotics and control tasks.
>
> **6. Theoretical guarantees:** We have the guarantee that the network is at least repaired for these samples, and, moreover, we have established experimentally that our repair generalizes better than retraining and fine-tuning. Note that retraining and fine-tuning sometimes cannot repair the NN even over the provided adversarial samples and that it is not easy to generate safe training samples from the adversarial samples. Finally, we proposed an algorithm in the supplementary materials, Section 4, that verifies the safety of repaired network.
>
> [1] Haarnoja, Tuomas, et al. "Learning to Walk Via Deep Reinforcement Learning." Robotics: Science and Systems. (2019).
>
> [r2] Sotoudeh, Matthew, and Aditya V. Thakur. "Provable repair of deep neural networks." Proceedings of the 42nd ACM SIGPLAN International Conference on Programming Language Design and Implementation. 2021.
>
> [r3] Fu, Feisi, and Wenchao Li. "Sound and Complete Neural Network Repair with Minimality and Locality Guarantees." International Conference on Learning Representations. 2021.